

# The impacts of vitamin D supplementation on serum levels of thyroid autoantibodies in patients with autoimmune thyroid disease: a meta-analysis

Dongdong Luo[1,2,*], Bojuan Li[1,*], Zhongyan Shan[1], Weiping Teng[1], Qigui Liu[3] and Jing Li[1]

[1] Department of Endocrinology and Metabolism, The Institute of Endocrinology, NHC Key Laboratory of Diagnosis and Treatment of Thyroid Disease, The First Hospital of China Medical University, Shenyang, Liaoning, China
[2] Department of Endocrinology and Metabolism, The Second Hospital of Dalian Medical University, Dalian, Liaoning, China
[3] School of Public Health, Dalian Medical University, Dalian, Liaoning, China
* These authors contributed equally to this work.

Corresponding author
Jing Li, lijingendocrine@126.com

## ABSTRACT

**Background:** Although vitamin D (VitD) deficiency had been found with close relationship with autoimmune thyroid disorders (AITD), the findings about the impacts of VitD supplementation on the production of anti-thyroid peroxidase (TPOAb) and anti-thyroglobulin antibodies (TgAb) remained inconsistent. Thus, a systemic meta-analysis was conducted to figure out the exact effects of VitD intervention on the production of TPOAb and TgAb in AITD patients.

**Methods:** We searched PubMed, Web of Science, Embase and The Cochrane Library databases for all clinical studies up to May 2023, which evaluated the changes in serum TPOAb and TgAb titers of AITD patients after VitD intervention. In addition, three subgroup analyses were carried out based on the duration of vitamin D supplementation, the initial nutritional status of participants, and the frequency of vitamin D administration.

**Results:** A total of 10 cohorts from 10 clinical trials with 577 patients with AITD were included. The serum titers of TPOAb and TgAb were significantly decreased after VitD supplementation as compared with their pre-treatment levels, respectively. However, a significant post-treatment reduction of serum TPOAb level was found only in the AITD patients with initial VitD insufficiency/deficiency, while an obvious decrease of serum TgAb was shown if only those studies without consideration of the pre-treatment VitD status were included. Serum TPOAb and TgAb titers were significantly decreased in the patients receiving VitD supplementation for at least 3 months, but not in the ones for less than 3 months. Serum titers of TPOAb and TgAb were both pronouncedly reduced in the patients receiving daily administration of VitD rather than weekly regimen. This study provides new evidence for the potential role of vitamin D supplementation in the treatment of AITD.

**Conclusions:** AITD patients may benefit with the reduction of TPOAb and TgAb production after a VitD supplementation (2800–60000 IU/week) for at least 3 months, especially with a decrease of serum TPOAb level when initially VitD insufficient/deficient, which mechanisms await further investigation.

**Systematic Review Registration:** PROSPERO, identifier CRD42023424029.

Anti-thyroglobulin antibody, Meta-analysis

# INTRODUCTION

Vitamin D (VitD) is currently considered as an important multifunctional hormone. The source of VitD in the body mainly relies on the endogenous production by skin epidermal cells under sunny ultraviolet radiation, and only a little comes from diet food. VitD insufficiency is very common in those people who are lack of sunshine exposure and nutrition intake. Serum 25(OH)D concentration is well-known to indicate the nutritional status of VitD with a level below 20 ng/mL as VitD deficiency and 21–29 ng/mL as VitD insufficiency (*Holick, 2007*). The representative isomers of VitD are ergocalciferol (VitD$_2$) and cholecalciferol (VitD$_3$), and the latter is the main form of VitD. The outcomes caused by VitD deficiency may be improved through direct supplementation with either VitD$_2$/VitD$_3$ or their active forms (*e.g.*, 1,25 (OH)$_2$D$_3$, 25(OH)D$_3$). The active compounds can modulate the activities of many organs, such as immune system and bone. The expression of the VitD receptor (VDR) is ubiquitous across all immune cells (*Chauss et al., 2022*; *Sassi, Tamone & D'Amelio, 2018*). It has been found that VitD may exert some crucial influences on both innate and adaptive immune response (*Bikle, 2009*). Several autoimmune diseases (*e.g.*, autoimmune diabetes mellitus, rheumatoid arthritis, and multiple sclerosis) have been found with close association with VitD deficiency (*Bellastella et al., 2015*; *Yamamoto & Jørgensen, 2019*). Clinical investigation and animal studies have suggested that VitD administration may inhibit the development of some autoimmune diseases (*Antico et al., 2012*; *Xie et al., 2017*).

Autoimmune thyroid diseases (AITD) are common autoimmune diseases, mainly consisting of Hashimoto's thyroiditis (HT), postpartum thyroiditis (PPT) and Graves' disease (GD). They are usually characterized by the presence of anti-thyroid peroxidase antibody (TPOAb) and/or anti-thyroglobulin antibody (TgAb). Given the well-known immunomodulatory effects of VitD, its association with AITD has been extensively investigated in recent years (*Zhao et al., 2021*). Cross-sectional studies have identified a correlation between VitD deficiency and the development of AITD (*Choi et al., 2014*; *Ke et al., 2017*; *Kim, 2016*; *Ma et al., 2015*; *Tamer et al., 2011*; *Unal et al., 2014*), which suggests the potential benefits of VitD supplementation in the AITD patients. There have been several articles published about the influence of VitD administration on the production of TPOAb and TgAb in the AITD patients. However, most of those human studies were small in the sample size, and the findings were inconsistent. Therefore, the aim of this study was to clarify the alterations in the serum levels of TPOAb and TgAb in the AITD patients following VitD intervention through a meta-analysis of the related

prospective studies, which may contribute to the most appropriate clinical application of VitD in the management of AITD.

## MATERIALS AND METHODS

### Search strategy

We conducted a systematic search of literature focusing on "Vitamin D" and "AITD" in the PubMed, EMBASE, Web of Science, and Cochrane Library databases up to May 2023. All studies on the changes of blood TPOAb and/or TgAb titers in AITD patients who were administered with VitD had been included into this analysis. Two reviewers (D.L. and B.L.) separately assessed the eligibility of the studies. The specific search strategies in the PubMed were as follows: ('Thyroiditis, Autoimmune' [Title/Abstract]) OR 'Autoimmune Thyroid*' [Title/Abstract] OR 'thyroiditis' [Title/Abstract] OR 'Hashimoto* Disease' [Title/Abstract] OR 'Hashimoto* Thyroiditi*' [Title/Abstract] OR 'Chronic Lymphocytic Thyroiditis' [Title/Abstract] OR 'Graves* Disease' [Title/Abstract] OR 'Basedows Disease' [Title/Abstract] OR 'Exophthalmic Goiters' [Title/Abstract] OR 'Postpartum Thyroiditis' [Title/Abstract] OR 'Thyroiditis, Autoimmune' [Mesh]) AND ('Vitamin D' [Mesh] OR 'vitamin D*' [Title/Abstract] OR 'Ergocalciferol' [Title/Abstract] OR 'Calciferol' [Title/Abstract] OR 'Cholecalciferol' [Title/Abstract] OR 'Dihydrotachysterol' [Title/Abstract] OR 'Calcifediol' [Title/Abstract] OR 'Calcitriol' [Title/Abstract] OR '1, 25-$(OH)_2D_3$' [Title/Abstract] OR '24,25-Dihydroxyvitamin $D_3$' [Title/Abstract] OR '25(OH)D' [Title/Abstract] OR '25OHD' [Title/Abstract]). Detailed search strategies of other databases above are listed in the File S1.

### Inclusion and exclusion criteria

The eligibility criteria were assembled using PICOS tool as follow (File S2): (1) population: patients diagnosed with AITD; (2) interventions: treated by VitD; (3) control: placebo or no treatment; (4) outcomes: the changes of serum TPOAb and/or TgAb levels after VitD treatment; (5) study type: any related prospective studies including randomized controlled trials (RCTs) published in scientific and peer-reviewed journals; (6) language: limited to English. No ethical consent was required because this study was performed based on those previously published data.

Exclusion Criteria: those studies in which AITD patients had received any other drugs (*e.g.*, selenium, iodine and inositol) than L-T4 and calcium were all excluded.

### Data extraction

Two independent investigators (D.L. and B.L.) retrieved the studies and any discrepancies were resolved by discussion with another reviewer (J.L.). Relevant data were extracted, including the first author, publication year and country, diseases investigated, study design, initial VitD status and its measurement, VitD administration regimen, sample size, sex, age, and serum levels of TPOAb, TgAb, thyrotropin, and 25(OH)D and their changes after

the intervention. The primary outcome of this meta-analysis was the effect of VitD supplementation on serum levels of thyroid autoantibodies in patients with AITD. The authors used an electronic data collection form to acquire the necessary information from each article.

## Quality assessment

Two reviewers completed a quality assessment to evaluate all included studies using the Cochrane Collaboration tool and the Methodological Index for Non-Randomized Studies (MINORS) criteria scoring system.

## Statistical analysis

Statistical analyses were conducted using Revman 5.3 software. Standardized mean difference (SMD) and 95% confidence intervals (CI) were used for continuous data. Heterogeneity was evaluated with $I^2$ statistics, with an $I^2$ value greater than 50% indicating significant heterogeneity. Forest plots were created to visually present the results of the meta-analysis, including effect sizes, confidence intervals, and the overall synthesis of findings across studies. Cochrane collaboration's tool and MINORS were used for assessing bias risk of RCT and non-RCT studies, respectively. A fixed-effects model was mainly implemented when $I^2$ was ≤50%, otherwise a random-effects model was adopted. Leave-one-out analyses were carried out to analyze the robustness when the heterogeneity was very high ($I^2 \geq 55\%$). $P$-value $< 0.05$ was considered as statistically different.

# RESULTS

## Characteristics of the studies included

During the screening process, 2,723 potentially relevant articles were obtained through database retrieval, and only nine studies were found eligible for the meta-analysis. Another study was identified from citation searching, and then was included. Finally, 10 studies were included into the current analysis, consisting of total 426 VitD-supplemented subjects and 151 controls (*Behera et al., 2020*; *Chahardoli et al., 2019*; *Chaudhary et al., 2016*; *Krysiak, Kowalcze & Okopien, 2016*; *Krysiak, Szkróbka & Okopień, 2017*, *2019*; *Mazokopakis et al., 2015*; *Simsek et al., 2016*; *Ucan et al., 2016*; *Vahabi Anaraki et al., 2017*). The screening process is illustrated in Fig. 1.

Ten studies utilized serum TPOAb and/or TgAb levels as the primary observation outcomes (Table 1). Among them, eight studies only enrolled HT patients (*Behera et al., 2020*; *Chahardoli et al., 2019*; *Chaudhary et al., 2016*; *Krysiak, Szkróbka & Okopień, 2017*, *2019*; *Mazokopakis et al., 2015*; *Ucan et al., 2016*; *Vahabi Anaraki et al., 2017*), one study included both HT and GD cases (*Simsek et al., 2016*), and one observed PPT women (*Krysiak, Kowalcze & Okopien, 2016*). Active forms of VitD were administered in one study (*Ucan et al., 2016*), while plain VitD was supplemented to AITD patients in the other nine studies. Among them, levothyroxine (L-T4) tablet was administered to the patients besides VitD supplementation in six studies (*Chahardoli et al., 2019*; *Chaudhary et al., 2016*; *Krysiak, Kowalcze & Okopien, 2016*; *Krysiak, Szkróbka & Okopień, 2017*;
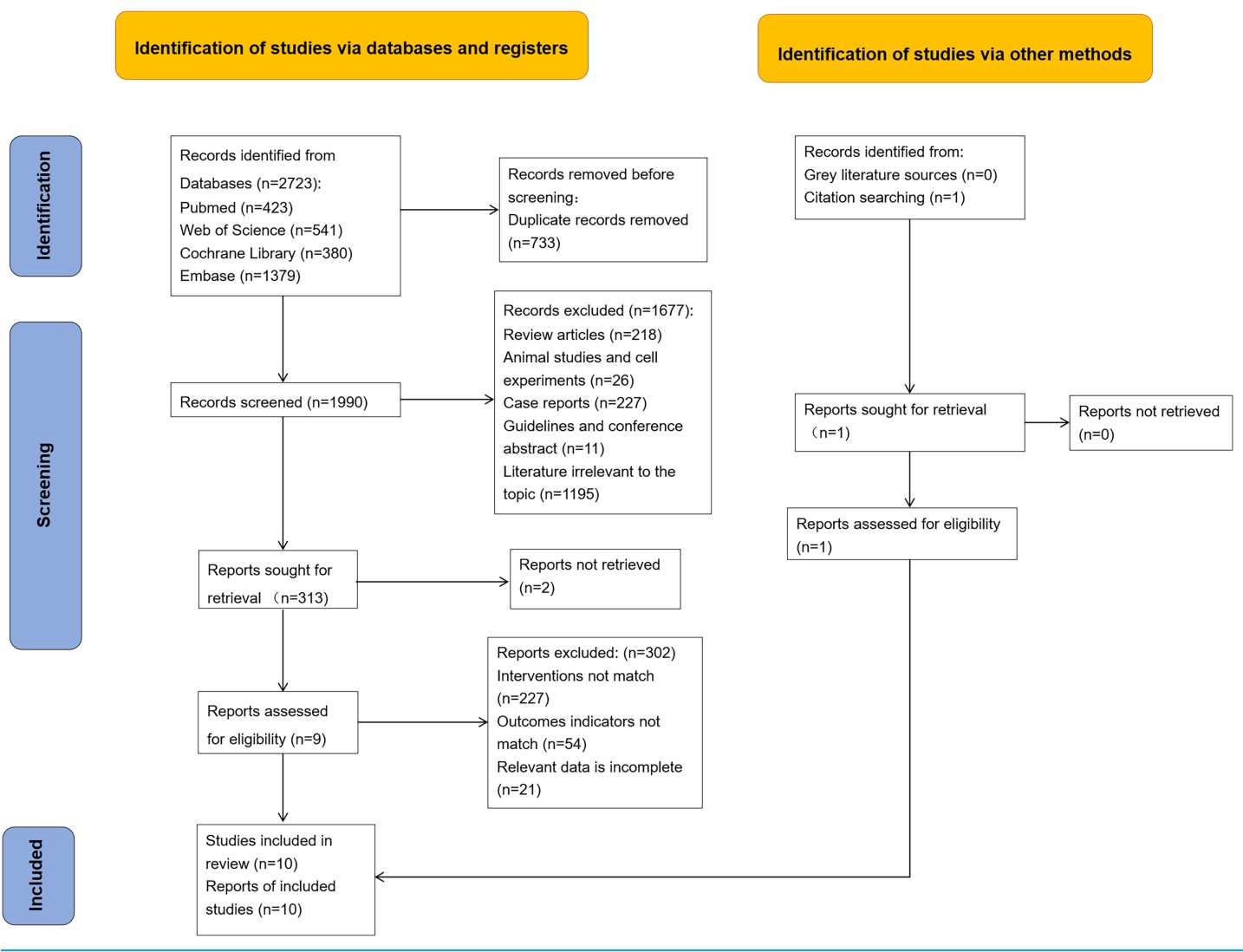

**Figure 1  Search flow diagram according to PRISMA guideline.**               

*Ucan et al., 2016*; *Vahabi Anaraki et al., 2017*), and calcium was additionally used in one study (*Chaudhary et al., 2016*). The results of the bias analysis are shown in Fig. 2. MINORS score of RCTs is in Table 2.

## Change in serum level of 25(OH)D after VitD intervention

Based on the data from the total 10 studies, the serum level of 25(OH)D exhibited a significant increase following VitD supplementation (SMD = 3.31, 95% CI [1.95–4.67], $P < 0.00001$, Fig. 3D), and its average level did not reach up to 30 ng/mL only in one study (Table 3). It indicated that the nutritional VitD statuses in those AITD patients were indeed improved after VitD supplementation through the reported regimens, although not all of them finally became VitD sufficient.

**Table 1 Characteristics of the 10 studies included in the meta-analysis.**

| First author, year | Country | Disease | Study design | VitD status | VitD measurement | VitD dose | Duration | Sample size (int./con.) | TPOAb in int. group (before vs. after) | TgAb in int. group (before vs. after) | Sex | Age (int./con.), years | Main outcome |
|---|---|---|---|---|---|---|---|---|---|---|---|---|---|
| *Mazokapakis et al. (2015)* | Greece | HT | Prospective | <30 ng/mL | CMIA | VitD$_3$ 1,200–4,000 IU/day | 4 months | n = 186 | Significant reduction | Reduction but NS | Both | 35.3 ± 8.5 | TPOAb, TgAb, BMI, TSH |
| *Ucan et al. (2016)* | Turkey | HT | Prospective | <20 ng/mL | RIA | 25(OH)D$_3$ 50,000 IU/week | 8 weeks | n = 25 | Significant reduction | Significant reduction | Both | 35.9 ± 11.2 | fT3, fT4, TSH, TPOAb, TgAb |
| *Simsek et al. (2016)* | Turkey | GD+HT | RCT | <20 ng/mL | LC–MS/MS | VitD 1,000 IU/day | 1 month | n = 82 (46/36) | Significant reduction | Significant reduction | Both | 35.8 ± 12/39.7 ± 12.6 | TSH, fT4, fT3, TPOAb, TgAb |
| *Chaudhary et al. (2016)* | India | AITD | RCT | No limited | RIA | VitD$_3$ 60,000 IU/week | 8 weeks | n = 100 (50/50) | Significant reduction | / | Both | 28.48 ± 6.57/27.86 ± 7.29 | fT4, TSH, Calcium, phosphate, iPTH |
| *Krysiak, Kowalcze & Okopień (2016)* | Poland | PPT | Prospective | <20 ng/mL | / | VitD 4,000 IU/day | 3 months | n = 11 | Significant reduction | Reduction but NS | Female | 32 ± 4 | fT3, fT4, TSH, Ca, phosphate, PTH |
| *Vahabi Anaraki et al. (2017)* | Iran | HT | RCT | <20 ng/mL | ELISA | VitD 50,000 IU/week | 12 weeks | n = 56 (30/26) | NS | / | Both | 43.55 ± 1.56/44.12 ± 1.56 | TPOAb, PTH, TSH |
| *Krysiak, Szkróbka & Okopień (2017)* | Poland | HT | RCT | ≥30 ng/mL | ELISA | VitD 2,000 IU/day | 6 months | n = 34 (16/18) | Significant reduction | Reduction but NS | Female | 34 ± 7/35 ± 6 | fT3, fT4, TSH, TPOAb, TgAb |
| *Krysiak, Szkróbka & Okopień (2019)* | Poland | HT | Prospective | Not limited | Competitive immunoassay | VitD 400 IU/day | 6 months | n = 20 | Significant reduction | Significant reduction | Male | 35 ± 8 | fT3, fT4, TSH, TPOAb, TgAb |
| *Chahardoli et al. (2019)* | Iran | HT | RCT | Not limited | ELISA | VitD$_3$ 50,000 IU/week | 3 months | n = 40 (19/21) | Reduction but NS | Significant reduction | Female | 36.4 ± 5.2/35.9 ± 7.8 | T3, T4, TSH, TPOAb, TgAb, Ca |
| *Behera et al. (2020)* | India | HT | Prospective | Not limited | CMIA | VitD$_3$ 60,000 IU/week | 6 months | n = 23 | Significant increase | / | Both | 35.5 ± 11.03 | fT4, TSH, TPOAb |

**Note:**
Abbreviations: CMIA, chemiluminescent microparticle immunoassay; con, control; ELISA, enzyme-linked immunoassay; fT3, free triiodothyronine; fT4, free thyroxine; GD, Graves' disease; HT, Hashimoto thyroiditis; int, intervention; iPTH, intact parathyroid hormone; LMS/MS, liquid chromatography-tandem mass spectrometry; NS, not statistically significant; PPT, postpartum thyroiditis; RCT, randomized controlled trial; TgAb, anti-thyroglobulin antibodies; TPOAb, anti-thyroid peroxidase antibodies; TSH, thyroid-stimulating hormone.

A

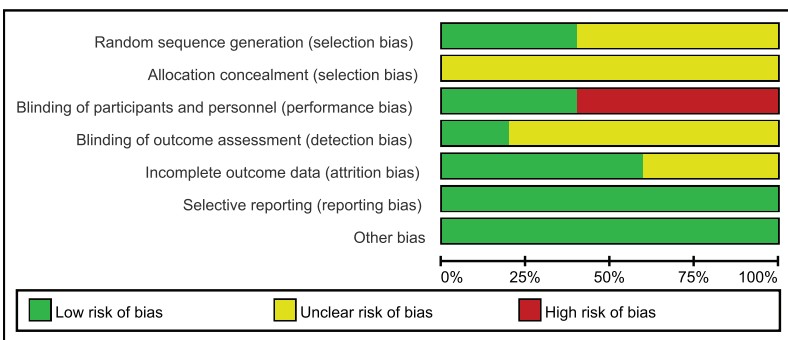

B

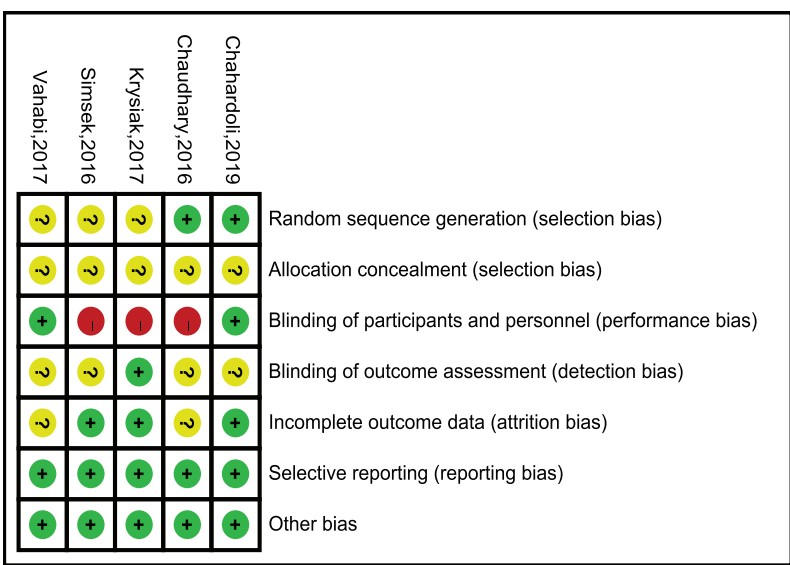

**Figure 2 Risk of bias assessments.** (A) Upper panel presents risk of bias for each included study; (B) Lower panel presents overall risk of bias of included studies; green indicates low risk, red indicates high risk, yellow indicates some concerns (*Chahardoli et al., 2019*; *Chaudhary et al., 2016*; *Krysiak, Szkróbka & Okopień, 2017*; *Simsek et al., 2016*; *Vahabi Anaraki et al., 2017*).

## Alterations in serum levels of TPOAb and TgAb after VitD intervention in all the 10 studies

In one study, the significant decrease in serum TPOAb level was not observed (*Vahabi Anaraki et al., 2017*), and even an obvious increase in serum TPOAb level after VitD supplementation was shown in another study (*Chaudhary et al., 2016*). However, the remaining eight studies had consistently exhibited the inhibitory effects of VitD intervention on the production of thyroid autoantibodies in the patients with AITD to some extents. Based on the data from the total 10 studies, serum TPOAb and TgAb titers were significantly reduced following the administration of VitD as compared with their baseline levels before VitD treatment, respectively (TPOAb: SMD = −0.44 95% CI [−0.73 to −0.15], $P = 0.003$, Fig. 3A; TgAb: SMD = −0.21, 95% CI [−0.36 to −0.05], $P = 0.009$,

**Table 2  Bias risk assessment of non-randomized trials included in this study by the MINORS scale.**

| Included studies | ① | ② | ③ | ④ | ⑤ | ⑥ | ⑦ | ⑧ | Additional criteria for comparative studies | | | | Total scores |
|---|---|---|---|---|---|---|---|---|---|---|---|---|---|
| | | | | | | | | | ⑨ | ⑩ | ⑪ | ⑫ | |
| *Mazokopakis et al. (2015)* | 2 | 2 | 1 | 2 | 0 | 2 | 2 | 0 | | | | | 11 |
| *Ucan et al. (2016)* | 2 | 2 | 1 | 2 | 0 | 2 | 2 | 0 | 1 | 1 | 2 | 2 | 17 |
| *Krysiak, Kowalcze & Okopien (2016)* | 2 | 2 | 1 | 2 | 0 | 2 | 2 | 0 | 2 | 2 | 2 | 2 | 19 |
| *Krysiak, Szkróbka & Okopień (2019)* | 2 | 2 | 2 | 2 | 2 | 2 | 2 | 0 | | | | | 14 |
| *Behera et al. (2020)* | 2 | 1 | 2 | 2 | 0 | 2 | 2 | 1 | | | | | 12 |

Note:
①Clearly stated aim; eInclusion of consecutive patients; ②Prospective collection of data; ③Endpoints appropriate to the aim of the study; ④Unbiased assessment of the study endpoint; ⑤Follow-up period appropriate to the aim of the study; ⑥Loss to follow up <5%; ⑦Prospective calculation of the study size; ⑧Adequate control group; ⑨Contemporary groups; ⑩Baseline equivalence of groups; ⑪Baseline equivalence of groups; ⑫Adequate statistical analyses. Abbreviation: MINORS, Methodological Index for Non-Randomized Studies.

Fig. 3B). No influence of VitD supplementation on TSH level was found (SMD = −0.14, 95% CI [−0.43 to 0.16], $P = 0.36$, Fig. 3C).

## Alterations in serum levels of TPOAb and TgAb after VitD intervention in only five RCTs

When only AITD patients from five RCTs were included into the analysis (*Chahardoli et al., 2019*; *Chaudhary et al., 2016*; *Krysiak, Szkróbka & Okopień, 2017*; *Simsek et al., 2016*; *Vahabi Anaraki et al., 2017*), a significant increase in serum VitD concentration (SMD = 3.20, 95% CI [1.59–4.81], $P < 0.0001$, Fig. 4D) and reduction in serum TPOAb titer (SMD = −0.24, 95% CI [−0.46 to −0.01], $P = 0.04$, Fig. 4A) in VitD-treated subjects were found as compared with those receiving only placebo. However, no significant difference was demonstrated in circulatory TgAb although it showed a consistent decreasing trend in all the three studies involved (SMD = −0.20, 95% CI [−0.52 to 0.12], $P = 0.22$, Fig. 4B). No influence of VitD supplementation on serum TSH level was found (SMD = 0.17, 95% CI [−0.54 to 0.88], $P = 0.64$, Fig. 4C).

## Alterations in serum levels of TPOAb and TgAb based on the duration of VitD treatment across all 10 studies

There were seven studies, in which total 305 AITD patients were treated with VitD for a duration of 3 months or longer (*Behera et al., 2020*; *Chahardoli et al., 2019*; *Krysiak, Kowalcze & Okopien, 2016*; *Krysiak, Szkróbka & Okopień, 2017*, *2019*; *Mazokopakis et al., 2015*; *Vahabi Anaraki et al., 2017*). Their serum TPOAb (SMD = −0.60, 95% CI [−1.01 to −0.18], $P = 0.005$, Fig. 5A) and TgAb titers (SMD = −0.25, 95% CI [−0.43 to −0.08], $P = 0.005$, Fig. 5B) were significantly decreased after VitD treatment as compared with the pre-treatment levels, respectively. In the other three investigations of total 121 subjects receiving VitD treatment for less than 3 months (*Chaudhary et al., 2016*; *Simsek et al., 2016*; *Ucan et al., 2016*), the post-treatment serum titer of either TPOAb (SMD = −0.18, 95% CI [−0.44 to 0.07], $P = 0.16$, Fig. 5C) or TgAb (SMD = −0.05, 95% CI [−0.38 to 0.28], $P = 0.77$, Fig. 5D) was not significantly decreased.

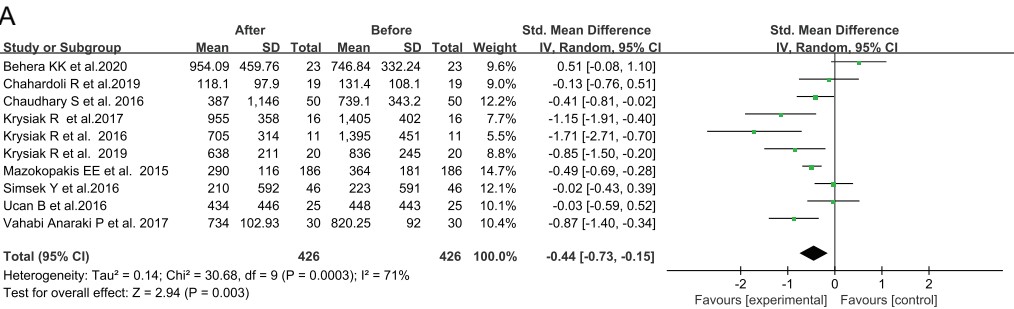

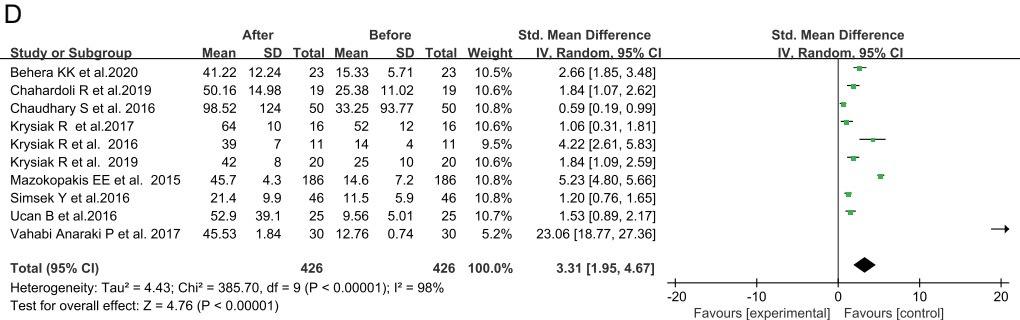

**Figure 3 Forest plots of serum thyroid autoantibodies, 25(OH)D and thyroid-stimulating hormone levels based on the whole 10 prospective studies.** (A) Anti-thyroid peroxidase antibody; (B) anti-thyroglobulin antibody; (C) thyroid-stimulating hormone; (D) 25-Hydroxyvitamin D (*Chahardoli et al., 2019*; *Chaudhary et al., 2016*; *Krysiak, Kowalcze & Okopien, 2016*; *Krysiak, Szkróbka & Okopień 2017, 2019*; *Ucan et al., 2016*; *Simsek et al., 2016*; *Vahabi Anaraki et al., 2017*; *Mazokopakis et al., 2015*; *Behera et al., 2020*).

**Table 3 The levels of serum thyroid autoantibodies, 25(OH)D and thyrotropin in the 10 studies.**

| First author, year | | TPOAb (IU/mL) | | TgAb (IU/mL) | | 25(OH)D (ng/mL) | | TSH (µIU/mL) | |
|---|---|---|---|---|---|---|---|---|---|
| | | **Before** | **After** | **Before** | **After** | **Before** | **After** | **Before** | **After** |
| *Mazokopakis et al. (2015)* | Int. | 364 ± 181 | 290 ± 116 | 16.8 ± 7.3 | 15.9 ± 5.4 | 14.6 ± 7.2 | 45.7 ± 4.3 | 2.5 ± 1.7 | 2.4 ± 1.5 |
| *Ucan et al. (2016)* | Int. | 448 ± 443 | 434 ± 446 | 365 ± 812 | 329 ± 831 | 9.56 ± 5.01 | 52.9 ± 39.1 | 3.20 ± 1.03 | 3.48 ± 1.49 |
| *Simsek et al. (2016)* | Int. | 223 ± 591 | 210 ± 592 | 312 ± 1,612 | 244 ± 915 | 11.5 ± 5.9 | 21.4 ± 9.9 | 4.1 ± 4.0 | 3.5 ± 2.5 |
| | Con. | 201 ± 593 | 166 ± 592 | 273 ± 566 | 237 ± 592 | 8.6 ± 4.2 | 10.9 ± 6.0 | 4.0 ± 2.5 | 3.5 ± 2.2 |
| *Chaudhary et al. (2016)* | Int. | 739.1 ± 343.2 | 387 ± 1,146 | ND | ND | 33.25 ± 93.77 | 98.52 ± 124 | 6.88 ± 138.98 | 3.16 ± 2.07 |
| | Con. | 687.8 ± 255.1 | 553.5 ± 1,002 | ND | ND | 39.61 ± 116.31 | 41.61 ± 100.06 | 6.80 ± 149.36 | 3.39 ± 2.19 |
| *Krysiak, Kowalcze & Okopien (2016)* | Int. | 1,395 ± 451 | 705 ± 314 | 1,410 ± 623 | 918 ± 456 | 14 ± 4 | 39 ± 7 | 2.5 ± 1.0 | 2.6 ± 1.2 |
| *Krysiak, Szkróbka & Okopień (2017)* | Int. | 1,405 ± 402 | 955 ± 358 | 1,210 ± 465 | 934 ± 415 | 52 ± 12 | 64 ± 10 | 4.3 ± 1.4 | 4.1 ± 1.5 |
| | Con. | 1,455 ± 390 | 1,410 ± 425 | 1,265 ± 420 | 1,212 ± 385 | 50 ± 10 | 47 ± 12 | 4.2 ± 1.3 | 4.5 ± 1.0 |
| *Vahabi Anaraki et al. (2017)* | Int. | 820.25 ± 92 | 734 ± 102.93 | ND | ND | 12.76 ± 0.74 | 45.53 ± 1.84 | 3.3 ± 0.5 | 3.88 ± 0.82 |
| | Con. | 838.07 ± 99.37 | 750.03 ± 108.71 | ND | ND | 13.28 ± 0.86 | 14.92 ± 1.06 | 3.45 ± 0.43 | 2.66 ± 0.38 |
| *Krysiak, Szkróbka & Okopień (2019)* | Int. | 836 ± 245 | 638 ± 211 | 756 ± 302 | 562 ± 267 | 25 ± 10 | 42 ± 8 | 2.9 ± 0.7 | 2.7 ± 0.7 |
| *Chahardoli et al. (2019)* | Int. | 131.4 ± 108 | 118.1 ± 97.9 | 192.6 ± 161.8 | 140.2 ± 134.3 | 25.38 ± 11.02 | 50.16 ± 14.98 | 3 ± 2.09 | 1.83 ± 1.4 |
| | Con. | 174.1 ± 141.8 | 181.6 ± 122.5 | 182.5 ± 153.9 | 176.7 ± 167.1 | 19.80 ± 8.81 | 22.03 ± 9.45 | 2.56 ± 1.36 | 2.77 ± 1.9 |
| *Behera et al. (2020)* | Int. | 746.84 ± 332.24 | 954.09 ± 459.76 | ND | ND | 15.33 ± 5.71 | 41.22 ± 12.24 | 7.23 ± 3.16 | 3.04 ± 2.62 |

**Note:**

Abbreviations: Con, control; Int, intervention; 25(OH)D, 25-hydroxyvitamin D; TgAb, antithyroglobulin antibodies; TPOAb, anti-thyroid peroxidase antibodies; TSH, thyroid-stimulating hormone; ND, not detected.

## Differential alterations in serum TPOAb and TgAb levels after VitD treatment with respect to the initial nutritional status of VitD in all the 10 studies

In five studies, total 276 AITD patients with initial serum 25 (OH) D below 30 ng/mL were enrolled, and supplemented with VitD (*Krysiak, Kowalcze & Okopien, 2016; Mazokopakis et al., 2015; Simsek et al., 2016; Ucan et al., 2016; Vahabi Anaraki et al., 2017*). Their post-treatment serum TPOAb but not TgAb titers were markedly decreased (SMD = −0.50, 95% CI [−0.89 to −0.10], $P = 0.01$, Fig. 6A), although the latter showed a

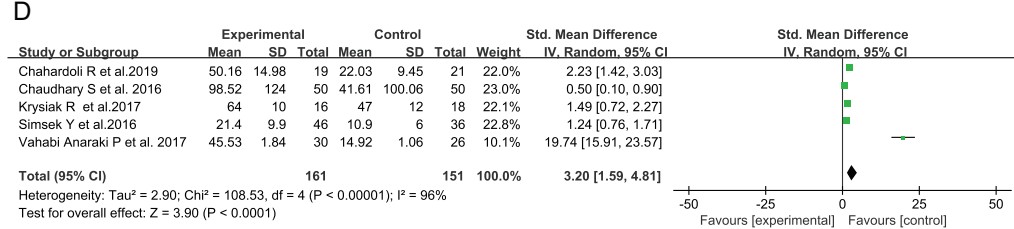

**Figure 4** **Forest plots of serum thyroid autoantibodies, 25(OH)D and thyroid-stimulating hormone levels based on the 5 randomized controlled trials.** (A) Anti-thyroid peroxidase antibody; (B) Anti-thyroglobulin antibody; (C) Thyroid-stimulating hormone; (D) 25-Hydroxyvitamin D (*Chahardoli et al., 2019*; *Krysiak, Szkróbka & Okopień, 2017*; *Simsek et al., 2016*; *Chaudhary et al., 2016*; *Vahabi Anaraki et al., 2017*).                

consistent decreasing trend (SMD = −0.14, 95% CI [−0.31 to 0.03], *P* = 0.10, Fig. 6B). In the other five investigations of total 140 patients (*Behera et al., 2020*; *Chahardoli et al., 2019*; *Chaudhary et al., 2016*; *Krysiak, Szkróbka & Okopień, 2017*, *2019*), their initial nutritional

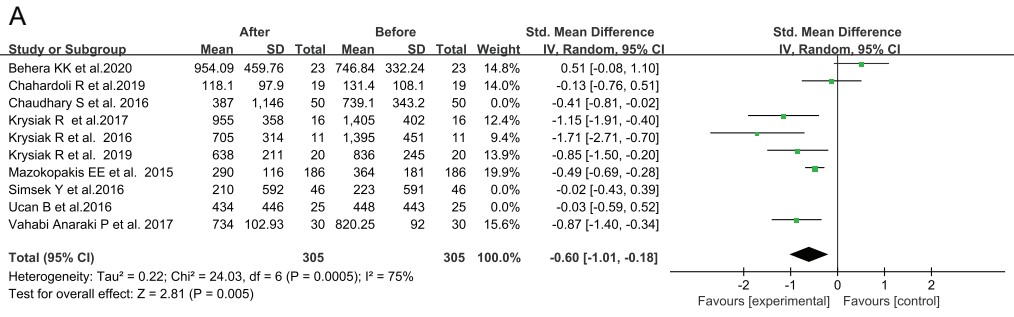

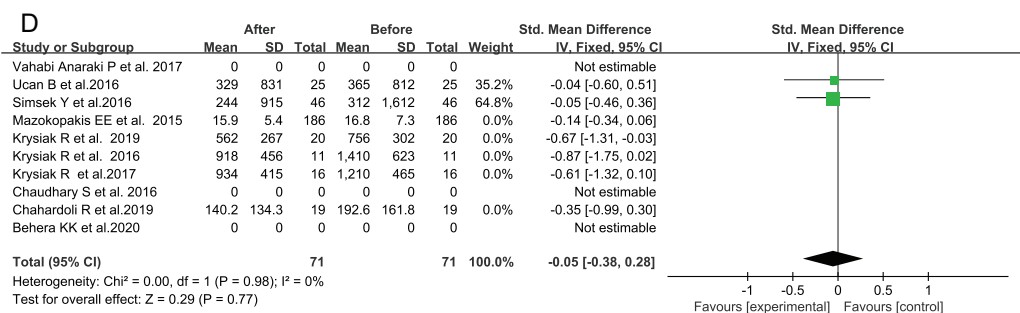

**Figure 5** **Forest plots of serum thyroid autoantibodies in the 10 prospective studies classified by the duration time of VitD supplementation.** (A) Anti-thyroid peroxidase antibody in AITD patients treated for at least 3 months; (B) anti-thyroglobulin antibody in AITD patients treated for at least 3 months; (C) anti-thyroid peroxidase antibody in AITD patients treated for less than 3 months; (D) anti-thyroglobulin antibody in AITD patients treated for less than 3 months (*Chahardoli et al., 2019*; *Chaudhary et al., 2016*; *Krysiak, Kowalcze & Okopien, 2016*; *Krysiak, Szkróbka & Okopień 2017*, *2019*; *Ucan et al., 2016*; *Simsek et al., 2016*; *Vahabi Anaraki et al., 2017*; *Mazokopakis et al., 2015*; *Behera et al., 2020*).

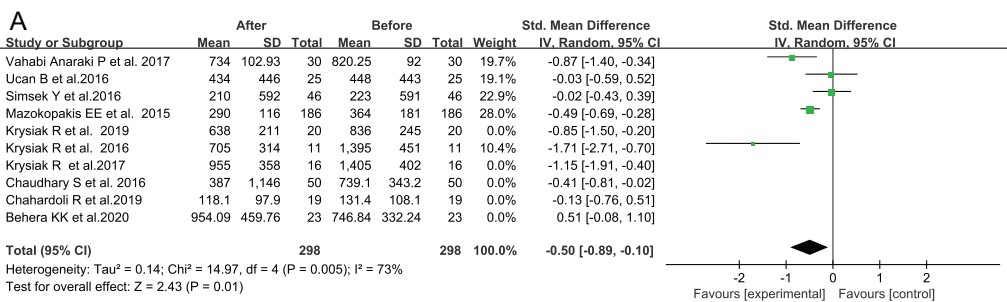

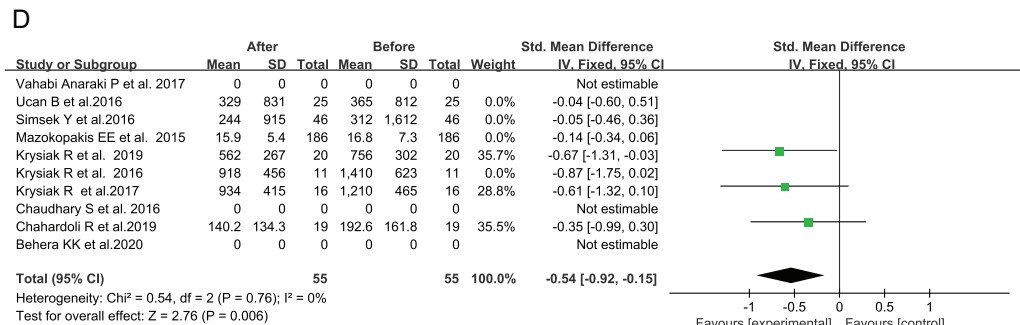

**Figure 6 Forest plots of serum thyroid autoantibodies in the 10 prospective studies classified by initial VitD nutritional statuses of the AITD patients.** (A) Anti-thyroid peroxidase antibody in treated AITD patients with initial serum 25(OH)D below 30 ng/ml; (B) anti-thyroglobulin antibody in the treated patients with initial serum 25(OH)D below 30 ng/ml; (C) anti-thyroid peroxidase antibody in treated AITD patients without initial VitD status considered; (D) anti-thyroglobulin antibody in treated AITD patients without initial VitD status considered (*Chahardoli et al., 2019*; *Chaudhary et al., 2016*; *Krysiak, Kowalcze & Okopien, 2016*; *Krysiak, Szkróbka & Okopień 2017, 2019*; *Ucan et al., 2016*; *Simsek et al., 2016*; *Vahabi Anaraki et al., 2017*; *Mazokopakis et al., 2015*; *Behera et al., 2020*).

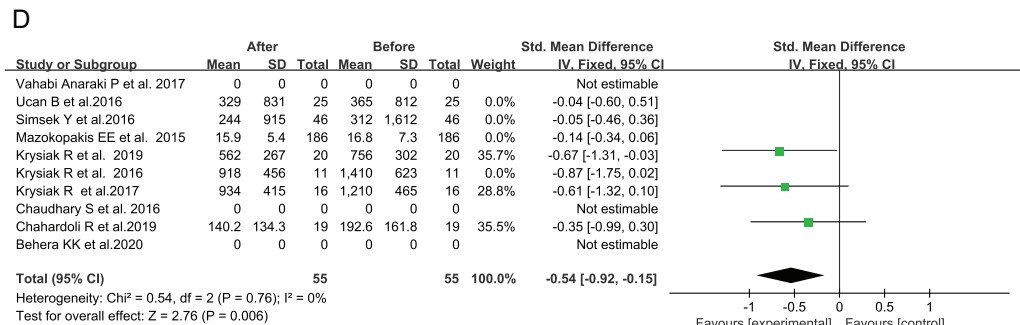

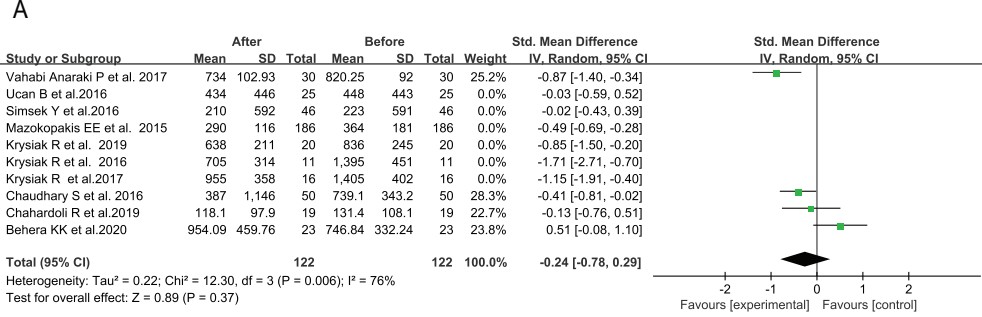

**A**

| Study or Subgroup | After Mean | SD | Total | Before Mean | SD | Total | Weight | Std. Mean Difference IV, Random, 95% CI |
|---|---|---|---|---|---|---|---|---|
| Vahabi Anaraki P et al. 2017 | 734 | 102.93 | 30 | 820.25 | 92 | 30 | 25.2% | -0.87 [-1.40, -0.34] |
| Ucan B et al.2016 | 434 | 446 | 25 | 448 | 443 | 25 | 0.0% | -0.03 [-0.59, 0.52] |
| Simsek Y et al.2016 | 210 | 592 | 46 | 223 | 591 | 46 | 0.0% | -0.02 [-0.43, 0.39] |
| Mazokopakis EE et al. 2015 | 290 | 116 | 186 | 364 | 181 | 186 | 0.0% | -0.49 [-0.69, -0.28] |
| Krysiak R et al. 2019 | 638 | 211 | 20 | 836 | 245 | 20 | 0.0% | -0.85 [-1.50, -0.20] |
| Krysiak R et al. 2016 | 705 | 314 | 11 | 1,395 | 451 | 11 | 0.0% | -1.71 [-2.71, -0.70] |
| Krysiak R et al.2017 | 955 | 358 | 16 | 1,405 | 402 | 16 | 0.0% | -1.15 [-1.91, -0.40] |
| Chaudhary S et al. 2016 | 387 | 1,146 | 50 | 739.1 | 343.2 | 50 | 28.3% | -0.41 [-0.81, -0.02] |
| Chahardoli R et al.2019 | 118.1 | 97.9 | 19 | 131.4 | 108.1 | 19 | 22.7% | -0.13 [-0.76, 0.51] |
| Behera KK et al.2020 | 954.09 | 459.76 | 23 | 746.84 | 332.24 | 23 | 23.8% | 0.51 [-0.08, 1.10] |
| **Total (95% CI)** | | | **122** | | | **122** | **100.0%** | **-0.24 [-0.78, 0.29]** |

Heterogeneity: Tau² = 0.22; Chi² = 12.30, df = 3 (P = 0.006); I² = 76%
Test for overall effect: Z = 0.89 (P = 0.37)

**B**

| Study or Subgroup | After Mean | SD | Total | Before Mean | SD | Total | Weight | Std. Mean Difference IV, Fixed, 95% CI |
|---|---|---|---|---|---|---|---|---|
| Vahabi Anaraki P et al. 2017 | 0 | 0 | 0 | 0 | 0 | 0 | | Not estimable |
| Ucan B et al.2016 | 329 | 831 | 25 | 365 | 812 | 25 | 0.0% | -0.04 [-0.60, 0.51] |
| Simsek Y et al.2016 | 244 | 915 | 46 | 312 | 1,612 | 46 | 0.0% | -0.05 [-0.46, 0.36] |
| Mazokopakis EE et al. 2015 | 15.9 | 5.4 | 186 | 16.8 | 7.3 | 186 | 0.0% | -0.14 [-0.34, 0.06] |
| Krysiak R et al. 2019 | 562 | 267 | 20 | 756 | 302 | 20 | 0.0% | -0.67 [-1.31, -0.03] |
| Krysiak R et al. 2016 | 918 | 456 | 11 | 1,410 | 623 | 11 | 0.0% | -0.87 [-1.75, 0.02] |
| Krysiak R et al.2017 | 934 | 415 | 16 | 1,210 | 465 | 16 | 0.0% | -0.61 [-1.32, 0.10] |
| Chaudhary S et al. 2016 | 0 | 0 | 0 | 0 | 0 | 0 | | Not estimable |
| Chahardoli R et al.2019 | 140.2 | 134.3 | 19 | 192.6 | 161.8 | 19 | 100.0% | -0.35 [-0.99, 0.30] |
| Behera KK et al.2020 | 0 | 0 | 0 | 0 | 0 | 0 | | Not estimable |
| **Total (95% CI)** | | | **19** | | | **19** | **100.0%** | **-0.35 [-0.99, 0.30]** |

Heterogeneity: Not applicable
Test for overall effect: Z = 1.05 (P = 0.29)

**C**

| Study or Subgroup | After Mean | SD | Total | Before Mean | SD | Total | Weight | Std. Mean Difference IV, Random, 95% CI |
|---|---|---|---|---|---|---|---|---|
| Vahabi Anaraki P et al. 2017 | 734 | 102.93 | 30 | 820.25 | 92 | 30 | 0.0% | -0.87 [-1.40, -0.34] |
| Ucan B et al.2016 | 434 | 446 | 25 | 448 | 443 | 25 | 17.1% | -0.03 [-0.59, 0.52] |
| Simsek Y et al.2016 | 210 | 592 | 46 | 223 | 591 | 46 | 20.4% | -0.02 [-0.43, 0.39] |
| Mazokopakis EE et al. 2015 | 290 | 116 | 186 | 364 | 181 | 186 | 24.6% | -0.49 [-0.69, -0.28] |
| Krysiak R et al. 2019 | 638 | 211 | 20 | 836 | 245 | 20 | 15.1% | -0.85 [-1.50, -0.20] |
| Krysiak R et al. 2016 | 705 | 314 | 11 | 1,395 | 451 | 11 | 9.5% | -1.71 [-2.71, -0.70] |
| Krysiak R et al.2017 | 955 | 358 | 16 | 1,405 | 402 | 16 | 13.2% | -1.15 [-1.91, -0.40] |
| Chaudhary S et al. 2016 | 387 | 1,146 | 50 | 739.1 | 343.2 | 50 | 0.0% | -0.41 [-0.81, -0.02] |
| Chahardoli R et al.2019 | 118.1 | 97.9 | 19 | 131.4 | 108.1 | 19 | 0.0% | -0.13 [-0.76, 0.51] |
| Behera KK et al.2020 | 954.09 | 459.76 | 23 | 746.84 | 332.24 | 23 | 0.0% | 0.51 [-0.08, 1.10] |
| **Total (95% CI)** | | | **304** | | | **304** | **100.0%** | **-0.57 [-0.96, -0.19]** |

Heterogeneity: Tau² = 0.15; Chi² = 17.31, df = 5 (P = 0.004); I² = 71%
Test for overall effect: Z = 2.90 (P = 0.004)

**D**

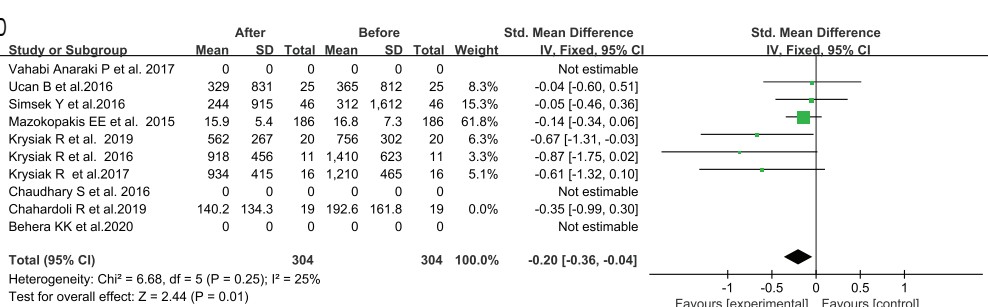

| Study or Subgroup | After Mean | SD | Total | Before Mean | SD | Total | Weight | Std. Mean Difference IV, Fixed, 95% CI |
|---|---|---|---|---|---|---|---|---|
| Vahabi Anaraki P et al. 2017 | 0 | 0 | 0 | 0 | 0 | 0 | | Not estimable |
| Ucan B et al.2016 | 329 | 831 | 25 | 365 | 812 | 25 | 8.3% | -0.04 [-0.60, 0.51] |
| Simsek Y et al.2016 | 244 | 915 | 46 | 312 | 1,612 | 46 | 15.3% | -0.05 [-0.46, 0.36] |
| Mazokopakis EE et al. 2015 | 15.9 | 5.4 | 186 | 16.8 | 7.3 | 186 | 61.8% | -0.14 [-0.34, 0.06] |
| Krysiak R et al. 2019 | 562 | 267 | 20 | 756 | 302 | 20 | 6.3% | -0.67 [-1.31, -0.03] |
| Krysiak R et al. 2016 | 918 | 456 | 11 | 1,410 | 623 | 11 | 3.3% | -0.87 [-1.75, 0.02] |
| Krysiak R et al.2017 | 934 | 415 | 16 | 1,210 | 465 | 16 | 5.1% | -0.61 [-1.32, 0.10] |
| Chaudhary S et al. 2016 | 0 | 0 | 0 | 0 | 0 | 0 | | Not estimable |
| Chahardoli R et al.2019 | 140.2 | 134.3 | 19 | 192.6 | 161.8 | 19 | 0.0% | -0.35 [-0.99, 0.30] |
| Behera KK et al.2020 | 0 | 0 | 0 | 0 | 0 | 0 | | Not estimable |
| **Total (95% CI)** | | | **304** | | | **304** | **100.0%** | **-0.20 [-0.36, -0.04]** |

Heterogeneity: Chi² = 6.68, df = 5 (P = 0.25); I² = 25%
Test for overall effect: Z = 2.44 (P = 0.01)

**Figure 7** **Forest plots of serum thyroid autoantibodies in the 10 prospective studies classified by VitD administration regimen.** (A) Anti-thyroid peroxidase antibody in AITD patients with weekly VitD supplementation; (B) anti-thyroglobulin antibody in AITD patients with weekly VitD supplementation; (C) anti-thyroid peroxidase antibody in AITD patients with daily VitD supplementation; (D) anti thyroglobulin antibody in AlTD patients with daily VitD supplementation (*Chahardoli et al., 2019*; *Chaudhary et al., 2016*; *Krysiak, Kowalcze & Okopien, 2016*; *Krysiak, Szkróbka & Okopień 2017, 2019*; *Ucan et al., 2016*; *Simsek et al., 2016*; *Vahabi Anaraki et al., 2017*; *Mazokopakis et al., 2015*; *Behera et al., 2020*).

status of VitD was not taken into the consideration, and even those subjects under sufficient VitD status were also enrolled and treated with VitD. Interestingly, no significant alteration was found in serum TPOAb titer (SMD = −0.38, 95% CI [−0.90 to 0.13], $P$ = 0.15, Fig. 6C), whereas post-treatment serum TgAb level was markedly decreased (SMD = −0.54, 95% CI [−0.92 to −0.15], $P$ = 0.006, Fig. 6D).

**Alterations in serum levels of TPOAb and TgAb based on the administration frequency of VitD across all 10 studies**

Among the four studies of total 112 AITD patients (*Behera et al., 2020*; *Chahardoli et al., 2019*; *Chaudhary et al., 2016*; *Vahabi Anaraki et al., 2017*), a weekly administration regimen was adopted. The post-treatment titers of serum TPOAb (SMD = −0.24, 95% CI [−0.78 to 0.29], $P$ = 0.37, Fig. 7A) and TgAb (SMD = −0.35, 95% CI [−0.99 to 0.30], $P$ = 0.29, Fig. 7B) were not significantly different from their pre-treatment levels, respectively. In the other six investigations of total 304 subjects with daily VitD supplementation (*Krysiak, Kowalcze & Okopien, 2016*; *Krysiak, Szkróbka & Okopień, 2017*, *2019*; *Mazokopakis et al., 2015*; *Simsek et al., 2016*; *Ucan et al., 2016*), the post-treatment serum TPOAb (SMD = −0.57, 95% CI [−0.96 to −0.19], $P$ = 0.004, Fig. 7C) and TgAb titers (SMD = −0.20, 95% CI [−0.36 to −0.04], $P$ = 0.01, Fig. 7D) were both markedly decreased.

Altogether, we found that continuous daily rather than weekly VitD supplementation for at least 3 months significantly reduced serum TPOAb level in the AITD patients with initial VitD insufficiency/deficiency, and also decreased serum TgAb in those patients potentially under VitD sufficient status. We had further done leave-one-out analyses on those results from highly heterogeneous studies ($I^2 \geq 55\%$), and confirmed our findings showed the relative robustness of the results except one subgroup (Fig. 6A) showed insufficient robustness, where VitD were given to intervene the serum TPOAb of those patients with initial serum 25 (OH) D level below 30 ng/mL (File S3).

## DISCUSSION

A series of clinical investigations and meta-analyses have demonstrated the association between VitD insufficiency/deficiency and increasing serum thyroid autoantibody levels in the patients with HT, GD and PPT, indicating the accelerating actions of VitD insufficiency/deficiency on the development of AITD (*Khozam et al., 2022*; *Štefanić & Tokić, 2020*; *Taheriniya et al., 2021*; *Wang et al., 2015*). The risks of GD, HT and PPT may be decreased by 1.55, 1.62 and 1.51 times, respectively, for every 5 nmol/l increase in serum 25 (OH) D concentration as indicated by several studies (*Ma et al., 2015*). Thus, the clinical application of VitD in the patients of AITD had received much attentions. However, since most of those investigations of the effects of VitD administration on the production of TPOAb and TgAb enrolled relatively small-size samples and they had not shown consistent findings. Therefore, a meta-analysis was further performed.

Indeed, there had been two meta-analysis publications about the effects of VitD intervention on the development of AITD yet, which included six (*Wang et al., 2018*) and seven (*Jiang et al., 2022*) studies, respectively. In the first meta-analysis, four RCTs published in English and two in Chinese were included, and 330 HT patients and 14 GD

ones were finally recruited. They found at least 6-month intervention of VitD could cause a significant decrease in serum TPOAb level, and both serum TPOAb and TgAb levels were markedly lower after VitD treatment than placebo administration. Except for the duration of VitD intervention, no other sub-group analyses were performed in the first meta-analysis (*Wang et al., 2018*). In the second meta-analysis of three RCTs and four prospective studies published before 2022, the findings based on the data from 258 HT patients showed that VitD supplementation significantly decreased serum TPOAb level as compared with that of the control group with either placebo or single use of levothyroxine, simvastatin or selenomethionine, but did not affect serum TgAb and thyroid functions (*Jiang et al., 2022*). They did subgroup-analyses of VitD intervention duration, pre-treatment VitD nutritional status and gender, and found that VitD intervention for at least 3 months reduced serum TPOAb level more than the control maneuver in both genders who were either VitD-insufficient or sufficient (*Jiang et al., 2022*). In the current analysis, we included five RCTs and five prospective studies published in English, totally consisting of 552 HT, 14 GD and 11 PPT patients. Those studies in which AITD patients in the VitD intervention group had received other drugs than L-T4 and calcium were all excluded from the current meta-analysis. Our results did not only show a significant reduction in serum TPOAb and TgAb levels after VitD supplementation based on all the 10 interventional studies, but also exhibit a markedly lower serum level of TPOAb after VitD intervention as compared with that of the control group based on the five RCTs. Furthermore, the current meta-analysis demonstrated the effectiveness of VitD supplementation in reducing both serum TPOAb level when VitD deficient/insufficient at the baseline and serum TgAb without respect to the initial serum VitD level. Finally, we found that daily other than weekly supplementation of VitD reduced serum TPOAb and TgAb levels, which had not been investigated in the previous meta-analysis studies.

Several studies have indicated that VitD may be an immunomodulatory hormone and exerts significant effects on the immune system due to the expression of VDR and 1α-hydroxylase (CYP27B1) in dendritic cells, macrophages, T cells, B cells, and other immune cells (*Ao, Kikuta & Ishii, 2021*; *Battault et al., 2013*; *Charoenngam & Holick, 2020*; *Provvedini et al., 1983*). The enzyme converts VitD into its active form of calcitriol, namely 1,25 $(OH)_2D_3$, which binds to VDR or PDIA3 on the target cells and exerts its effects through both genomic and non-genomic mechanisms (*Chichiarelli et al., 2022*; *Pol et al., 2022*). A recent randomized controlled trial known as VITAL, involving 25,871 American participants, demonstrated that daily supplementation of 2000 IU of VitD for a continuous period of 5 years can lead to a significant reduction in the risk of developing any autoimmune disease including autoimmune thyroid disorders by 22% (*Hahn et al., 2022*). It had been found that the supplementation of 1,25 $(OH)_2D_3$ to the experimental autoimmune thyroiditis (EAT) animal ameliorated the pathological changes of the thyroid gland, inhibited thyroid auto-antibody production and corrected the cytokine disequilibrium (*Wang et al., 2025*; *Zhang et al., 2024*).

Our study revealed significant decreases in serum TPOAb and TgAb titers following VitD supplementation. However, the subgroup analysis showed that VitD

supplementation could reduce serum TPOAb level when the patients were VitD deficient/insufficient at the baseline whereas it decreased serum TgAb when those patients with potentially VitD sufficient were recruited. It suggests that higher serum VitD concentration may be needed to suppress the production of TgAb. TPOAb is well known as the most specific biomarker for the presence of thyroid autoimmunity, representing a kind of adaptive immune response. It is expressed in about 95% of HT patients with significant correlation with the degree of intrathyroidal lymphocyte infiltration (*Antonelli et al., 2015*). TgAb is existent in only 60–80% of HT patients with less sensitivity and specificity for AITD diagnosis, reflecting an innate type of immune response (*McLachlan & Rapoport, 2004*). It has previously reported that there is a poor but significant correlation between TPOAb and TgAb expressions (*Caturegli, De Remigis & Rose, 2014*). VitD has been known as a threshold nutrient, and its immunomodulatory actions may not be linearly correlated with its serum concentration (*Caturegli, De Remigis & Rose, 2014*). It has been found that vitamin D may exert differential modulatory effects on innate immune reactions and adaptive immune functions (*Johnson & Thacher, 2023*). Specifically, $1,25(OH)_2D_3$ enhances the innate immune response while acting as a suppressor of adaptive immunity. Thus, we speculated that the different production mechanisms between TPOAb and TgAb may account for their differential changes in AITD patients after VitD intervention, and the serum threshold level for VitD to suppress the production of TgAb may be higher than that of TPOAb. In addition, the previous study of EAT has shown that VitD treatment inhibited the production of TgAb in the mice without VitD deficiency (*Zhang et al., 2024*).

In order to figure out the appropriate regimen of VitD administration for AITD patients, we further performed the subgroup analyses on the duration and dosing frequency. Our results indicate that daily other than weekly VitD supplementation for at least 3 months can reduce serum levels of TPOAb and TgAb. We admitted that there were some limitations in this meta-analysis. There was still some heterogeneity in the included studies in this meta-analysis, which was attributed to the limited overall number of studies, small sample sizes in those studies, and their differential VitD administration regimens, initial VitD nutritional status, baseline thyroid functions and sex ratios of participants. Some studies combined VitD supplementation with L-T4 treatment while others did not. Although we have further done leave-one-out analyses on those results from highly heterogeneous studies, and confirmed our preliminary findings (Table 3), this meta-analysis could not exclude all the heterogeneity factors among the studies. Besides, no subgroup analysis was performed based on plain and active forms of VitD supplementation due to limited data. Thus, more large-size sample RCTs are still needed to clarify the benefits of VitD supplementation and the optimal dosing regimen in AITD patients.

## CONCLUSIONS

In conclusion, our current meta-analysis findings suggest that VitD supplementation for at least 3 months can lower serum TPOAb and TgAb levels in AITD patients. Mainly AITD

patients with VitD deficiency/insufficiency can benefit with the decrease of serum TPOAb from VitD supplementation. But VitD may be administered to AITD patients even with VitD sufficiency so as to reduce serum TgAb. The frequency for VitD supplementation on a daily basis may be more effective than a weekly regimen. Our findings have further provided clinical evidence for the optimal administration regimen of VitD in AITD patients. However, the optimal range of 25(OH)D level to achieve the best protection against AITD, and the optimal duration, dosing regimen, intervention form (plain *vs.* active) of VitD to ensure the safety and maximize the benefits still await more investigations in large-size sample RCTs.

### Funding
This work was supported by Beijing Medical Award Foundation (grant number YXJL-2024-0350-0089), National Ministry of Science and Technology, National Key Research and Development Plan "Common Disease Prevention Research" key project, (grant number 2023YFC2508303), Liaoning Province, Basic Research Project, (grant number No. LJ212410159030) to Jing Li. The funders had no role in study design, data collection and analysis, decision to publish, or preparation of the manuscript.

### Grant Disclosures
The following grant information was disclosed by the authors:
Beijing Medical Award Foundation: YXJL-2024-0350-0089.
National Ministry of Science and Technology.
National Key Research and Development Plan "Common Disease Prevention Research" key project: 2023YFC2508303.
Liaoning Province, Basic Research Project: LJ212410159030.

### Competing Interests
The authors declare that they have no competing interests.

### Author Contributions
- Dongdong Luo conceived and designed the experiments, analyzed the data, prepared figures and/or tables, authored or reviewed drafts of the article, and approved the final draft.
- Bojuan Li performed the experiments, prepared figures and/or tables, authored or reviewed drafts of the article, and approved the final draft.
- Zhongyan Shan conceived and designed the experiments, authored or reviewed drafts of the article, resources, and approved the final draft.
- Weiping Teng conceived and designed the experiments, authored or reviewed drafts of the article, resources, and approved the final draft.
- Qigui Liu analyzed the data, authored or reviewed drafts of the article, and approved the final draft.

- Jing Li conceived and designed the experiments, authored or reviewed drafts of the article, and approved the final draft.

## Data Availability

This is a systematic review/meta-analysis.

## Supplemental Information

Supplemental information for this article can be found online at http://dx.doi.org/10.7717/peerj.19541#supplemental-information.

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
