# Peer review of "The impacts of vitamin D supplementation on serum levels of thyroid autoantibodies in patients with autoimmune thyroid disease: a meta-analysis"

_PeerJ, doi:10.7717/peerj.19541_

## Round 0.1 · original submission · Major Revisions

Please respond to the reviewers' point by point.

·

Basic reporting

The manuscript titled “The impacts of vitamin D supplementation on serum levels of thyroid autoantibodies in patients with autoimmune thyroid disease: a meta-analysis” was a meta-analysis study to clarify the alterations in the serum levels of TPOAb and TgAb in the AITD patients following VitD intervention. In this study, the authors conducted a thorough comprehensive literature search of PubMed, EMBASE, Web of Science, and Cochrane Library databases and finally enclosed 10 studies. After information extraction and synthesis, they found that after VitD administration serum TPOAb and TgAb titers were significantly reduced. The subgroup analysis is reasonable with detailed content and a reliable conclusion. The writing language of the manuscript is smooth and easy to understand, and the expression is clear. The results of this meta-analysis can provide evidence on whether vitamin D supplementation is necessary for AITD and helps in making clinical treatment decisions.

Experimental design

The following issues still need to be mentioned:
1. Regarding the SEARCH STRATEGY (Line 84-95), The specific search strategies expressed are not correct。 It should be like: ”(…OR…OR…) AND (…OR…OR…)”, the authors should pay attention to the item between “AND” and “OR”, parentheses cannot be omitted.

2. The resolution of all the Figures (FIG. 1-7) does not meet the requirements, and it is very difficult to see the contents of the pictures clearly

Validity of the findings

-

Additional comments

-

·

Basic reporting

This manuscript deals with an interesting topic related to vitamin D supplementation and AITD patients. It has a fair level of background and context, and the text is well-written.

However, the authors provide a group of images with inferior quality; thus, this manuscript must be improved before its publication at PeerJ.

Experimental design

This manuscript has a well-defined research question and also is relevant and meaningful. However, the authors need to describe their methods with enough detail and information to replicate their statistical analysis, then the manuscript needs to be improved.

Validity of the findings

The conclusion of this manuscript is well-stated and linked to the original research question.

Additional comments

Detailed comments and suggestions are in the attached file; please revise them. I will be honored to revise the revised version of this manuscript.

---

## Round 0.2 · accepted · Accept

All the comments were addressed.

·

Basic reporting

The revised manuscript titled “The impacts of vitamin D supplementation on serum levels of thyroid autoantibodies in patients with autoimmune thyroid disease: a meta-analysis” was a meta-analysis study to clarify the alterations in the serum levels of TPOAb and TgAb in the AITD patients following VitD intervention. In this study, the authors conducted a though comprehensive literature search of PubMed, EMBASE, Web of Science, and Cochrane Library databases and finally enclosed 10 studies. After information extraction and synthesis, they found that after VitD administration serum TPOAb and TgAb titers were significantly reduced. The subgroup analysis is reasonable with detailed content and reliable conclusion. The results of this meta-analysis can provide evidence on whether vitamin D supplementation is necessary of AITD and help making clinical treatment decisions.
All the comments raised before have been properly revised or explained.

Experimental design

no comment

Validity of the findings

no comment

Additional comments

Line133-134 ...forest plots were created to visually reflect the heterogeneity degree among those studies included. This statement is not correct. The forest plot is a comprehensive display of the results of meta-analysis,NOT “reflect the heterogeneity degree among those studies”

·

Basic reporting

The authors have addressed all comments and suggestions well. But now it needs a very profound proofreading to make it suitable for publication at PeerJ

Experimental design

The authors have addressed all comments and suggestions well. But now it needs a very profound proofreading to make it suitable for publication at PeerJ

Validity of the findings

The authors have addressed all comments and suggestions well. But now it needs a very profound proofreading to make it suitable for publication at PeerJ